# Understanding GANs via Generalization Analysis for Disconnected Support

## Abstract

This paper provides theoretical analysis of generative adversarial networks (GANs) to explain its advantages over other standard methods of learning probability measures. GANs learn a probability through observations, using the objective function with a generator and a discriminator. While many empirical results indicate that GANs can generate realistic samples, the reason for such successful performance remains unelucidated. This paper focuses the situation where the target probability measure satisfies the *disconnected support property*, which means a separate support of a probability, and relates it with the advantage of GANs. It is theoretically shown that, unlike other popular models, GANs do not suffer from the decrease of generalization performance caused by the disconnected support property. We rigorously quantify the generalization performance of GANs of a given architecture, and compare it with the performance of the other models. Based on the theory, we also provide a guideline for selecting deep network architecture for GANs. We demonstrate some numerical examples which support our results.

## 1 Introduction

*Generative Adversarial Networks* (GANs) (Goodfellow et al., 2014) attract much attention as technology for learning a distribution and generating data. The purpose of GANs is to learn a probability measure from a given dataset and generate samples from the learned measure. It is often seen that samples generated by GANs can extract effectively features in the real world; it is difficult, for instance, to distinguish real images and generated images. By practical successes, a countless number of variations of GANs have been developed (Dziugaite et al., 2015; Arjovsky et al., 2017; Li et al., 2015; Nowozin et al., 2016; Gulrajani et al., 2017; Zhao et al., 2016), and applied to a wide range of tasks (Reed et al., 2016; Zhu et al., 2017; Gauthier, 2014).

Understanding the remarkable performance of GANs is, however, still a challenging problem. There are active discussions on the role of generators in its learning scheme (Goodfellow, 2016; Arjovsky & Bottou, 2017; Arora et al., 2018; Creswell et al., 2018), and adversarial structures with discriminators are also a target of interest (Lotter et al., 2015; Zhang et al., 2018). A gaming structure between generators and discriminators is also regarded as a useful factor in the mechanism of GANs (Mescheder et al., 2017; Arora et al., 2017; Heusel et al., 2017). Generalization performance of GANs has been investigated in several studies (Liang, 2017; Liu et al., 2017; Tolstikhin et al., 2017). In spite of these studies, it is not yet clear why GANs can generate well-extracted data better than other standard methods.

This paper introduces the *disconnected support property*, and explains an advantage of GANs in connection with this notion. The disconnected support property refers to a probability measures of which the support is divided into several disjoint sets, allowing non-differentiable density on the boundary. The property makes a probability measure be complex, hence it can be an obstacle for standard methods to learn the measure effectively. This property, however, is popularly seen in many real data, especially data with cluster structure, as demonstrated in Section 3.

We investigate in detail the approximation and estimation ability of GANs and some other methods, and provide novel generalization analysis of probability measures with disconnected support. Firstly, we show that the other methods suffer worse generalization performance due to complex structures of disconnected supports (Proposition 1 and Lemma 2). Secondly, our generalization analysis reveals that GANs can learn the probability measure without loss of efficiency under the the disconnected

supports (Theorem 1, Corollary 1 and **??**). Additionally, we derive a guideline for choosing the number of layers or connections of the generator and discriminator from the generalization analysis. Numerical results support our theoretical findings.

We remark that the disconnected support property is different from the low-dimensional supports studied in Arjovsky & Bottou (2017), where the support of a measure generated by neural networks is disjoint to the measure of observations. In contrast, this paper considers the case in which the support of the observation measure is divided into disjoint subsets. The problem of disconnected supports is complement to the low-dimensionality, hence these two problems can be investigated separately. In this paper, to simplify the discussion, we assume that the support of a probability measure is not low-dimensional.

The contributions of this paper are summarized as follows:

1. We show that GANs perform better than other standard methods of estimating probability measures when the measure satisfies the disconnected support property.
2. We provide a new generalization error bound under a general formulation of GANs by analyzing an approximation error. The result is thus applicable to a wide range of variations of GANs.
3. Based on the generalization bound, we provide a theoretical guideline for selecting architectures of generators and discriminators.

All the proofs are given in Supplementary materials.

## 2 PRELIMINARIES

### 2.1 NOTATION

We use notations $I := [0, 1]$. A $j$-th element of a vector $b$ is denoted by $b_j$, and $\|b\|_q := (\sum_j b_j^q)^{1/q}$ is the $q$-norm ($q \in [0, \infty]$). $\mathrm{vec}(\cdot)$ is a vectorization operator for matrices. For $z \in \mathbb{N}$, $[z] := \{1, 2, \ldots, z\}$ is a set of positive integers no more than $z$. For $\alpha \in \mathbb{R}$, $\lfloor \alpha \rfloor$ denotes the largest integer which is not larger than $\alpha$. For a domain $\Omega$ in a Euclidean space and a function $f : \Omega \to \mathbb{R}$, $\|f\|_{L^p} := (\int_\Omega |f(t)|^p dt)^{1/p}$ denotes the $L^p$ norm for $p \in [0, \infty]$. For $f : \Omega \to \mathbb{R}^D$ with a multi-dimensional output, $f_d$ denotes a $d$-th coordinate of $f(x) = (f_1(x), \ldots, f_D(x))^\top$. Let $H^\beta(\Omega)$ be the Hölder space for $\beta > 0$ such as a set of $\beta$-smooth functions $f : \Omega \to \mathbb{R}$, namely, $f$ is $C^{\lfloor \beta \rfloor}$-class and its $\lfloor \beta \rfloor$-th derivative is $\beta - \lfloor \beta \rfloor$-Hölder continuous. $\otimes$ denotes a tensor product, and $\circ$ a composition of functions, namely, for functions $f$ and $f'$, $f \circ f' = f(f'(\cdot))$. A Borel $\sigma$-algebra of $\Omega$ is denoted as $\sigma(\Omega)$. For a measurable mapping $f : \Omega \to \Omega'$ and $B' \subset \Omega'$, a pre-image of $f$ is defined as $f^{-1}(B') := \{t \in \Omega \mid B' \ni f(t)\}$. Let $\boldsymbol{I}_\Omega : x \mapsto \{0, 1\}$ be an indicator function such that $\boldsymbol{I}_\Omega(x) = 1$ if $x \in \Omega$, and $\boldsymbol{I}_\Omega(x) = 0$ otherwise.

### 2.2 GENERAL FRAMEWORK OF GANS

We provide a general formulation of a learning problem with *generative adversarial networks (GANs)* following Liu et al. (2017). In this paper, we consider a probability measure $P^*$ on a measurable space $(I^D, \Sigma)$ with a dimensionality $D \in \mathbb{N}$ and $\Sigma := \sigma(I^D)$. Here, we set $D \geqslant 3$. Suppose we have a set of $n$ observations $\mathcal{D} := \{X_1, \ldots, X_n\}$ which is independently and identically generated from $P^*$. Let $P_n := \frac{1}{n} \sum_{i \in [n]} \delta_{X_i}$ be an empirical measure where $\delta_x$ is the Dirac measure at $x$.

The goal of generative networks is to estimate $P^*$ from $\mathcal{D}$. To this end, we construct a probability measure by *generators*. Let $P_Z$ be the uniform distribution on $(I^D, \Sigma)$. For a measurable mapping $g : I^D \to I^D$, we define $P_g$ as the pushforward measure: i.e., $P_g(B) = P_Z(g^{-1}(B))$ for $B \in \Sigma$. We call $g$ as a generator and use $\mathcal{G}$ for a set of generators.

GANs employ a learning scheme with a metric with *discriminators* (Goodfellow et al., 2014). Let $\mathcal{F} = \{f : I^D \to \mathbb{R}\}$ be a a set of discriminators. This paper considers a general metric for GANs (Liu et al., 2017),

$$d_\mathcal{F}(P, P') := \sup_{f \in \mathcal{F}} \mathbb{E}_{X \sim P}[f(X)] - \mathbb{E}_{X \sim P'}[f(X)], \tag{1}$$

between probability measures $P$ and $P'$.

For the learning process, we generate $m$ noise samples $\widetilde{Z}_1, ..., \widetilde{Z}_m$ from $P_Z$ and obtain generated samples as $\widetilde{X}_j := g(\widetilde{Z}_j)$ with $g \in \mathcal{G}$ for $j \in [m]$. Let $P_{g,m} := \frac{1}{m} \sum_{j \in [m]} \delta_{\widetilde{X}_j}$ denote the sampling measure. GANs construct an estimator $P_{\widehat{g}}$ for P* by learning $\widehat{g}$ with the following optimization problem

$$\widehat{g} \in \underset{g \in \mathcal{G}}{\operatorname{argmin}} \, d_{\mathcal{F}} \left( P_n, P_{g,m} \right). \tag{2}$$

The metric (1) covers a wide variety of GANs by selecting $\mathcal{F}$. Among others, the original GAN (Goodfellow et al., 2014) is realized if $\mathcal{F}$ contains a logarithm of density ratio; Wasserstein-GAN (Arjovsky et al., 2017), MMD-GAN (Dziugaite et al., 2015; Li et al., 2017) and Energy-Based GAN (Zhao et al., 2016) are given if $\mathcal{F}$ is the set of 1-Lipschitz functions, a reproducing kernel Hilbert space, and the bounded continuous functions, respectively. The $f$-GAN (Nowozin et al., 2016) also belongs to this class.

We assume that $\mathcal{F}$ is large enough to contain functions which can work as a discriminator, namely, we assume that the following holds:

$$d_{\mathcal{F}}(P, P') = 0 \Leftrightarrow P = P'. \tag{3}$$

A sufficient condition for (3) is investigated in Zhang et al. (2018).

## 2.3 Deep Neural Networks for Generators and Discriminators

In the schemes of GANs, $\mathcal{F}$ and $\mathcal{G}$ are realized by *deep neural networks* (DNNs). For further discussion, we formulate the function class given by DNNs.

Let $L \in \mathbb{N}$ be a number of layers in DNNs, and $D'_\ell \in \mathbb{N}$ be a dimensionality of variables in an $\ell$-th layer for $\ell \in [L + 1]$. Here, we set $D'_{L+1} = D$ for generators and $D'_{L+1} = 1$ for discriminators. We introduce $A_\ell \in \mathbb{R}^{D'_{\ell+1} \times D'_\ell}$ and $b_\ell \in \mathbb{R}^{D'_\ell}$ as matrix and vector parametersof the $\ell$-th layer. An *architecture* $\Theta$ of DNNs is defined as a set of $L$ pairs of $(A_\ell, b_\ell)$ as $\Theta := ((A_1, b_1), ..., (A_L, b_L))$. We define notations for $\Theta$ as follow: $|\Theta| := L$ as the number of layers, $\|\Theta\|_0 := \sum_{\ell \in [L]} \| \operatorname{vec}(A_\ell)\|_0 + \|b_\ell\|_0$ as the number of non-zero elements in $\Theta$, and $\|\Theta\|_\infty := \max\{\max_{\ell \in [L]} \| \operatorname{vec}(A_\ell)\|_\infty, \max_{\ell \in [L]} \|b_\ell\|_\infty\}$ be the scale of parameters in $\Theta$. We employ the ReLU activation function $\eta : \mathbb{R}^{D'} \to \mathbb{R}^{D'}$ for each $D' \in \mathbb{N}$ such as $\eta(x) = (\max\{x_d, 0\})_{d \in [D']}$.

We define functions of DNNs with an architecture $\Theta$ as $\xi[\Theta] : \mathbb{R}^{D'} \to \mathbb{R}^{D''}$ by

$$\xi[\Theta](x) = x^{(L+1)}, \; x^{(1)} := x, \; x^{(\ell+1)} := \eta(A_\ell x^{(\ell)} + b_\ell), \text{ for } \ell \in [L].$$

The function class of DNNs is thus given by

$$\Xi(S, B, L) := \left\{ \xi[\Theta] : I^D \to \mathbb{R} \mid \|\Theta\|_0 \leqslant S, \|\Theta\|_\infty \leqslant B, |\Theta| \leqslant L \right\},$$

where $S \in \mathbb{N}$, $B > 0$, and $L \in \mathbb{N}$ are hyper-parameters. Here, $S$ bounds the number of non-zero parameters of DNNs, namely, it controls the sparseness of DNNs. $B$ is a bound for scales of parameters.

## 3 Disconnected Support Property

### 3.1 Introduction and Example

It is often observed that data in real world data the support of its probability measure may not be connected but a union of disjoint subsets. This is typical if the data has cluster structures, as seen in many data sets for classification tasks. Moreover, the density function of the probability measure may not be smooth at a boundary of the support. Figures 1 (MNIST, (LeCun et al., 1998)) and 2 (Shelter Animal, Center) illustrate such examples in the real world. They are projected onto a 2-dimensional Euclidean space by t-SNE (Maaten & Hinton, 2008) so that they preserve the original distance structure among points. We can see that both of the data are concentrated on several disjoint

subsets and there are a clear gap or empty regions between some of the subsets. This observation suggests that the disconnected property of probability measures should be addressed in discussing estimation of probability measures, while standard analysis does not consider this phenomenon. In fact, this paper will show that the disconnected supports property has an important role in showing an advantage of GANs over standard estimation methods.

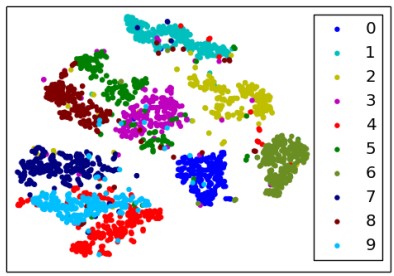

Figure 1: Plot of the MNIST data.

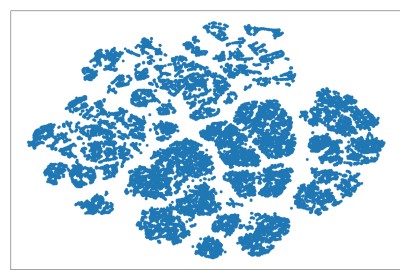

Figure 2: Plot of the animal data.

### 3.2 MATHEMATICAL FORMULATION OF DISCONNECTED SUPPORTS

Here we make a rigorous definition of disconnected supports. The property of *smoothness* (i.e. differentiability) is involved, which is a key factor to analyze generalization performance in the fields of the statistics (Stone, 1982; Tsybakov, 2009); Stone (1982) shows, for instance, that smoothness and a dimension of data are sufficient to characterize an optimal convergence of generalization errors.

We first prepare a family of subsets as a component in the disconnected supports:

$$\mathcal{S}_{\alpha,J} := \left\{ S \subset I^D \mid \text{A boundary of } S \text{ is } J \text{ combination of } \alpha\text{-smooth hyper surfaces} \right\}.$$

The supplementary material will provide a more rigorous definition.

Now, we define the disconnected support property of probability measures as well as a probability measure with global support, i.e., with no the disconnected support property. Let $\text{Supp}(P)$ be the support of $P$., i.e, $\text{Supp}(P) := \{x \in I^D \mid P(V_x) > 0 \text{ for all open neighborhood } V_x \text{ of } x\}$. Hereafter, $M \geq 2$ is the number of disjoint components of a support.

**Definition 1.** (Disconnected Supports / Global Support)
Let $M \geq 2$. A probability measure $P$ on $(I^D, \Sigma)$ has $M$ *disconnected supports*, if there exist nonempty disjoint sets $S_1, ..., S_M \in \mathcal{S}_{\alpha,J}$ such that

$$\text{Supp}(P) = \bigcup_{m \in [M]} S_m.$$

A probability measure $P$ on $(I^D, \Sigma)$ has a *global support*, if $\text{Supp}(P) = I^D$.

Figure 3 illustrates the disconnected support property.

We next formulate a notion of smoothness for $P$ with disconnected supports. Let $\beta \geq 1$ be a parameter for a degree of smoothness of $P$.

**Definition 2.** (Local Smoothness)
A probability measure $P$ with $M$ disconnected support is *locally $\beta$-smooth*, if there exist $M$ pairs $(\widetilde{S}_m, S_m) \in \mathcal{S}_{2\beta,J} \times \mathcal{S}_{2\beta,J}$ and $\beta + 1$-smooth bijective measurable maps $\gamma_m : \widetilde{S}_m \to S_m$ as

$$P(B) = P_Z(\gamma_m^{-1}(B)), \forall B \in \sigma(S_m),$$

for $m \in [M]$.

This definition of local smoothness says that a probability measure $P$ with disconnected supports can be generated by sufficiently smooth mappings $\gamma_m$. It is used for considering a smooth density function of $P$ restricted on $S_m$.

**Lemma 1.** *(Locally Smooth Density Functions)*
*If a probability measure $P$ with $M$ disconnected support is locally $\beta$-smooth, then there exists a function $p_m : S_m \to \mathbb{R}_+$ such that*

$$P(B) = \int_B p_m(x)d\lambda, B \in \sigma(S_m),$$

*where $\lambda$ is the Lebesgue measure, and $p_m$ is $\beta$-smooth for all $m \in [M]$.*

We call $p_m$ as a *local density function*.

Note that, since $P$ with disconnected supports is not absolutely continuous to the Lebesgue measure on $I^D \backslash \bigcup_{m \in [M]} S_m$, an ordinary density function cannot be defined. Instead, a localized version of density functions for each $S_m$ is introduced, which is guaranteed by the local smoothness.

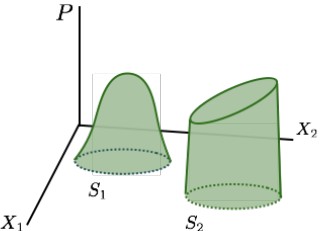
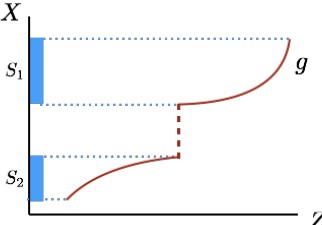

Figure 3: Illustration of a probability measure $P$ with a disconnected support. $\mathrm{Supp}(P)$ is a union of two disjoint sets $S_1$ and $S_2$.

Figure 4: Illustration of a generator $g$. To represent discontinuous $S_1$ and $S_2$, $g$ should be discontinuous.

### 3.3 DIFFICULTY WITH DISCONNECTED SUPPORTS

As shown in this subsection, the generalization performance of many standard estimation methods is worsened with disconnected supports. We consider popular nonparametric methods, for which the generalization performance is well-studied in the asymptotics of the observation size $n$. The considered methods are the kernel density estimator (KDE) (Nadaraya, 1964), the nonparametric Bayes (NB) by the Dirichlet mixtures of normal distributions (Ferguson, 1973), the series density estimator (SDE) (Efromovich et al., 2008; Efromovich, 2010) and the density estimator with Gaussian process (GP) (Leonard, 1978). Bounds of the generalization errors for these methods are already known (Tsybakov, 2009; Ghosal et al., 2007; van der Vaart & van Zanten, 2008), and they are optimal in the minimax sense. Here, their performance is evaluated with respect to a root of an expected squared loss with respect to the $L^2$-norm, namely, $d_2(P, P') := \mathbb{E}[\|p - p'\|_{L^2}^2]^{1/2}$ where $p$ and $p'$ are densities for $P$ and $P'$.

We show the deterioration of their performance by the disconnected support property. Let $\widehat{P}$ be an estimator for $P^*$ by KDE, NB, SDE, or GP. If $P^*$ has a global support and a $\beta$-smooth density, the existing studies (Tsybakov, 2009; Ghosal et al., 2007; van der Vaart & van Zanten, 2008) show that

$$d_2(P^*, \widehat{P}) = O\left(n^{-\beta/(2\beta+D)}\right).$$

These bounds are sufficiently tight, since these bounds corresponds to an optimal rate (Stone, 1982), and the performance of the methods can be improved as the density for $P^*$ is smoother (larger $\beta$).

On the other hand, we consider a case in which $P^*$ has the disconnected support property.

**Proposition 1.** *(Deterioration of other standard methods)*
*There exists $P^*$ of the disconnected support property and locally $\beta$-smooth such that*

$$d_2(P^*, \widehat{P}) = O\left(n^{-1/(2+D)}\right).$$

When $P^*$ has disconnected supports, the errors are worse than those for the global support, independent of $\beta$. This worse generalization error can be understood by the non-smoothness or discontinuity of the density functions on the boundaries of the disconnected sets (see Figure 3).

We next discuss other generative models for estimating a probability measure. To the best of our knowledge, other probabilistic generative methods (Koller et al., 2009) and the variational autoencoder (Kingma & Welling, 2013), their statistical generalization property is not well investigated. Here we provide a property of generators for probability measures with disconnected supports.

**Lemma 2.** *(Discontinuous Generators for Disconnected Supports)*
*If $P^*$ has disconnected supports and $P^* = P_{g*}$ with a generator $g^*$, then $g^*$ is not uniformly continuous.*

Lemma 2 states that a generator must be discontinuous to construct a probability measure with disconnected support sets. Intuitively, to make $P_{g*}(B) = 0$ for $B \in I^D$ with $\lambda(B) > 0$, the slope of $g^*$ at $z \in I^D$ should be close to infinite for $z \in g^{*,-1}(B)$, hence $g^*$ cannot be uniformly continuous (see Figure 4). Because of the discontinuity, generative models with smooth functions, such as an adversarial generative model with kernel generators (Sinn & Rawat, 2018), cannot work well with disconnected supports.

## 4 GENERALIZATION BY GANS

We provide generalization analysis for GANs for probability measures with and without disconnected supports. For the purpose, we employ a metric $d_{\mathcal{F}}$ with properly selected discriminators $\mathcal{F}$ and evaluate the generalization error $d_{\mathcal{F}}(P^*, P_{\hat{g}})$ with respect to an observation size $n$ and a sampling size $m$. We assume $\mathcal{F}$ is realized by DNNs as $\mathcal{F} = \Xi(S_f, B_f, L_f) \cap \widetilde{\mathcal{F}}$ with parameters $S_f, B_f, L_f$, where $\widetilde{\mathcal{F}}$ is a specified functional class; for an example, $\widetilde{\mathcal{F}}$ is 1-Lipschitz functions for Wasserstein-GAN. Here, we consider settings that all $f \in \mathcal{F}$ are $L_1$-Lipschitz continuous and $\|f\|_{L^\infty} \leqslant B_F$ with constants $L_1, B_F > 0$. Generators are also constructed by DNNs as $\mathcal{G} = \Xi(S_g, B_g, L_g)$ with parameters $S_g, B_g, L_g$.

A standard line of discussing generalization, we should consider statistical errors and approximation errors. We define a measure of the complexity of $\mathcal{F}$

$$\Upsilon_n(\mathcal{F}) := \inf_{\eta > 0} 4\eta + 12n^{-1/2} \int_\eta^c \log \mathcal{N}((L_1 + 1)^{-1}\epsilon, \mathcal{F}, \|\cdot\|_n)^{1/2} d\epsilon,$$

where $c > 0$ is a constant depends on $\mathcal{F}$ and $\mathcal{N}(\epsilon, \widetilde{\mathcal{F}}, \|\cdot\|)$ is a covering number of $\widetilde{\mathcal{F}}$ with respect to an empirical norm $\|\cdot\|$. We note that $\Upsilon_n(\mathcal{F})$ bounds an expectation of the Rademacher complexity as

$$\Upsilon_n(\mathcal{F}) \geqslant \mathbb{E}\left[\sup_{f \in \mathcal{F}} \frac{1}{n}\left|\sum_{i \in [n]} \tau_i f(X_i)\right|\right],$$

where $\tau_i$ is the i.i.d. Rademacher random variables; $\Pr(\tau_i = 1) = \Pr(\tau_i = 1) = 1/2$, and the expectation is about $X_i$ and $\tau_i$. Using the statistics and learning theory van der Vaart & Wellner (1996); Bartlett et al. (2005), we can apply a bound for $\Upsilon_n(\mathcal{F})$ as

$$\Upsilon_n(\mathcal{F}) \leqslant C_{\mathcal{F}} n^{-1/\kappa},$$

with some constants $C_{\mathcal{F}} > 0$ and $\kappa \geqslant 2$.

Regarding approximation errors, we need to consider approximation of a discontinuous function; since Lemma 2 shows that a discontinuous generator is necessary to represent disconnected supports. To approximate such generators, DNNs in GANs has an advantage.

**Lemma 3.** *(Approximation for Discontinuous $g$ by DNNs)*
*Suppose $P^*$ has $M$ disconnected supports and locally $\beta$-smooth, and also $P^* = P_{g*}$ holds with some $g^*$. Then, for any $S_g$, there exist $\mathcal{G}$, $\dot{g} \in \mathcal{G}$, and a constant $c_g = c_g(B_g, L_g) > 0$ such that*

$$\|\dot{g}_d - g_d^*\|_{L^2} \leqslant c_g M S_g^{-\beta/D}, \ \forall d \in [D]. \tag{4}$$

*Furthermore, if $P^* = P_{g*}$ has a global support and it is $\beta$-smooth, (4) holds with $M = 1$.*

Lemma 3 shows that $\mathcal{G}$ for GANs can approximate $g^*$ for disconnected supports with the rate $(-\beta/D)$ by $S_g$, and the rate is same in the case of global support. This implies an advantage of GANs in comparison with the other standard methods (Proposition 1).

Based on Lemma 3, we obtain the main theorem for generalization analysis.

**Theorem 1.** *(Generalization of GANs)*
*Suppose that $P^*$ has $M$ disconnected supports and locally $\beta$-smooth, and we have $n$ observations and $m$ samplings. Then, with $\mathcal{F}$, an existing $\mathcal{G}$, an estimator $P_{\hat{g}}$ by (2), and finite constants $c_1 = c_1(L_f, B_f, L_g, B_g), c_2, c_3 = c_3(L_f, B_f) > 0$, the following inequality holds with high probability,*

$$d_{\mathcal{F}}(P^*, P_{\hat{g}}) \leqslant \underbrace{\Upsilon_m(\widetilde{\mathcal{F}}) + c_1 \frac{\sqrt{S_g} + \sqrt{S_f}}{\sqrt{m}}}_{=:I} + \underbrace{c_2 M D S_g^{-\beta/D}}_{=:II} + \underbrace{\Upsilon_n(\widetilde{\mathcal{F}}) + c_3 \sqrt{\frac{S_f}{n}}}_{=:III}. \qquad (5)$$

*Furthermore, if $P^*$ has a global support and it is $\beta$-smooth, (5) holds with $M = 1$.*

Each of the terms $I, II$ and $III$ has the following role: $I$ bounds an error by the $m$ samplings, $II$ bounds an error from approximation by $\mathcal{G}$, and $III$ bounds an error by $n$ observations.

**Proof Outline**: By the definition of $P_{\hat{g}}$ in (2) and standard calculation, we obtain the inequality

$$d_{\mathcal{F}}(P^*, P_{\hat{g}}) \leqslant \underbrace{2 \sup_{g \in \mathcal{G}} \sup_{f \in \mathcal{F}} \left| \mathbb{E}_{P_{g,m}}[f(X)] - \mathbb{E}_{P_g}[f(X)] \right|}_{=:i} + \underbrace{\inf_{g \in \mathcal{G}} d_{\mathcal{F}}(P_g, P^*)}_{=:ii} + \underbrace{2 d_{\mathcal{F}}(P_n, P_0)}_{=:iii}.$$

To obtain $i \leqslant I$ and $iii \leqslant III$, we apply an empirical process technique (van der Vaart & Wellner, 1996), especially convergence of integral probability measures (Sriperumbudur et al., 2012) and the entropy control technique (Lemma 4 and 5 in the supplementary material). To show $ii \leqslant II$, we employ recent results on approximation ability of DNNs (Yarotsky, 2017; Petersen & Voigtlaender, 2017; Imaizumi & Fukumizu, 2018) and obtain an approximation bound for $d_{\mathcal{F}}(P_g, P^*)$ (Lemma 3). Combining these results, we obtain the statement of Theorem 1. □

Theorem 1 provides two trade-off relations with respect to $S_g$ and $S_f$. The generator class $\mathcal{G}$ controls uncertainty by sampling and the approximation error, while $S_f$ controls uncertainty of observations and discrimination. For balancing the trade-offs, we select the number of parameters (connections of DNNs) with some constants $c_g, c_f > 0$ as

$$S_g = c_g m^{D/(2\beta+D)}, \text{ and } S_f = c_f n^{(\kappa-2)/\kappa}, \qquad (6)$$

for optimizing the bound (5). We then obtain the following corollary.

**Corollary 1.** *(Convergence Rate of GANs)*
*Make the same assumptions as Theorem 1, and set $S_f$ and $S_g$ as in (6). Then, with high probability converging to 1, we obtain*

$$d_{\mathcal{F}}(P^*, P_{\hat{g}}) = O\left(n^{-1/\kappa}\right) + O\left(m^{-1/\kappa} + m^{-\beta/(2\beta+D)}\right). \qquad (7)$$

A selection of $\widetilde{\mathcal{F}}$ determines the first term in (7), since $\kappa$ depends on $\widetilde{\mathcal{F}}$. For an example, when $\mathcal{F}$ is a set of 1-Lipschitz functions, the first term is $O\left(n^{-1/(2+2D)}\right)$ (Sriperumbudur et al., 2012).

**Remark 1.** (Heterogeneous Smoothness)
Corollary 1 can be extended when $P^*$ has different smoothness for each $m \in [M]$, i.e., $P^*$ is locally $\beta_m$-smooth on a set $S_m$. In this case, we can easily extend our analysis in Theorem 1 and Corollary 1, and obtain the following convergence rate.

$$d_{\mathcal{F}}(P^*, P_{\hat{g}}) = O\left(n^{-1/\kappa}\right) + O\left(m^{-1/\kappa} + m^{-\tilde{\beta}/(2\tilde{\beta}+D)}\right),$$

where $\tilde{\beta} := \min_{m \in [M]} \beta_m$.

## 5 DISCUSSION

We emphasize the theoretical results show that GANs do not suffer from the effect of disconnected supports. A larger $\beta$ improves performance of GANs even with disconnected supports, as shown

in Theorem 1 and Corollary 1. This phenomenon is different from the result of the other methods discussed in Proposition 1. Hence, we can state that GANs have advantages over the other methods which are deteriorated by the disconnected property (Section 3.3). In other words, when data are generated from a probability measure with disconnected supports and sufficiently smooth in each of the sets, only GANs can estimate the measure effectively and the other methdos cannot. This advantage of GANs comes from the approximation power for discontinuous generators shown in Lemma 3.

The results (5) and (7) provide interpretation about performance of GANs. About convergence with $n$, the complexity of $\widetilde{\mathcal{F}}$ and $S_f$ control a trade-off between convergence and a power of discrimination. While smaller $S_f$ reduce the errors in terms of $d_{\mathcal{F}}$, too small $S_f$ can lose the power of discrimination to satisfy (1). Hence, setting $S_f$ as in (6) can keep the discrimination power and does not worsen the overall rate of convergence $O(n^{-1/\kappa})$. About convergence with $m$, $S_g$ controls the trade-off between the bias and variance of the estimator. An optimal way to select $S_g$ is provided in (6) which depends on $\beta$ and $D$, and it is more important when $\kappa$ is small (e.g. $\kappa = 2$ as MMD-GAN). Based on the interpretation and the selection rule (6), our study can provide a guideline for a design of the architecture of DNNs.

## 5.1 RELATED WORKS

Compared with studies for understanding GANs (Dziugaite et al., 2015; Liang, 2017; Liu et al., 2017; Zhang et al., 2018; Biau et al., 2018; Arora et al., 2017), we show that GANs can avoid influences of disconnected supports, unlike the other methods, hence it is a source of the advantage of GANs. This paper is the first work to focus on the disconnected support property, while several discussions (Goodfellow, 2016; Liang, 2017; Creswell et al., 2018) focus on models and metrics of the learning scheme of GANs.

It is important to compare our result with other studies for generalization analysis. Although some existing studies (Dziugaite et al., 2015; Liang, 2017; Liu et al., 2017; Zhang et al., 2018; Biau et al., 2018; Arora et al., 2017) provide generalization analysis, they do not analyze an approximation effect, namely, they evaluate $d_{\mathcal{F}}(P^*, P_{\hat{g}}) - \inf_{g \in \mathcal{G}}(P^*, P_g)$. Since we analyze the term $\inf_{g \in \mathcal{G}}(P^*, P_g)$, we can provide a more general bound and discuss the effect of disconnected support.

As we mentioned in the introduction, this paper is not along with studies for low-dimensional supports (Arjovsky & Bottou, 2017). We also note that an optimization aspect and gaming aspect of GANs are out of concerns of this paper. We focus on the statistical aspects of GANs such as sample complexity.

## 6 NUMERICAL EXPERIMENTS

We compare the numerical performance of GANs and the other methods with toy data with. We generate synthetic data from the following two settings: (A) Gaussian distribution restricted on a compact set (global support), and (B) a probability measure with two disconnected supports (density function is the black solid line in Figure 6). We generate $n = 500, 1000, ..., 5000$ observations and estimate the true probability measures with Wasserstein GAN, MMD-GAN, KDE (the Gaussian kernel and the Epanechnikov kernel), SDE (Fourier basis), and NB (Dirichlet process prior). Hyperparameters for these methods are selected by cross-validation. For GANs, we set $m = n$. We use $d_{\mathcal{F}}$ to evaluate errors by GANs, and a root of the expected squared errors with the $L^2$-norm for the other methods. The plots are the mean of 30 replications.

Figure 5 shows generalization errors by the methods. With the disconnected support case (B), we plot the estimated density in Figure 6. The black line shows the true density, the dashed line is by estimated densities of the other methods, and bars are histograms by GANs. The results by Wasserstein-GAN and MMD-GAN are almost same, so we omit the result by Wasserstein-GAN.

In Figure 5, we see that in the case of global support (A), the error of GANs and the other methods are comparable. In contrast, in the case of disconnected supports (B), the other standard methods show worse generalization and only GANs keep the high performance. From Figure 6, we can see that GANs can reveal the disconnected supports, while some of the other methods fail to fit. KDE with the Gaussian kernel represents the disconnected support by employing a small bandwidth. However, the small bandwidth yields a too sharp density, tending to worsen the generalization performance.

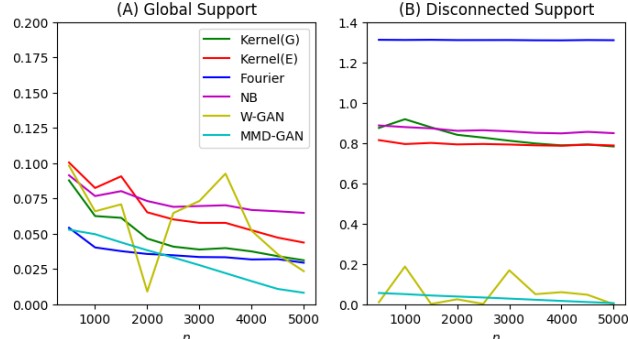 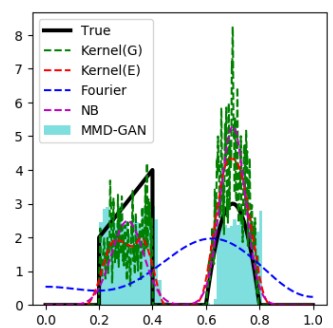

Figure 6: Estimated density functions with the case (B).

Figure 5: Generalization errors.

## 7 CONCLUSION

We investigate a generalization performance of GANs with a situation such that a support of real probability measures is divided into several sets. We find that GANs do not suffer from the division of supports, while some of the other nonparametric methods loss their efficiency by the division. Since real data are often distributed on such divided supports, the finding in this paper is related to the question of why GANs perform well with real datasets.

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

# Supplementary Materials for "Understanding GANs via disconnected Support Detection"

We introduce a new notation $Pf := \mathbb{E}_{X \sim P}[f(X)]$ with a probability measure $P$ and a function $f$. For a set $\Omega$ with equipped distance $d$, let $\mathcal{N}(\epsilon, \Omega, d)$ be a covering number which is a minimum number of $\epsilon$-balls to cover $\Omega$.

## A  SOME ADDITIONAL INFORMATION

### A Rigorous Definition of $\mathcal{S}_{\alpha, J}$

We consider a set represented by a combination of multiple *horizon functions*, which has been used in Petersen & Voigtlaender (2017). Given $\alpha$-smooth function $h \in H^\alpha(I^{D-1})$ with $\alpha \geqslant 1$, a horizon function $\Psi_h : I^D \to \{0, 1\}$ is defined for some $d \in [D]$ as

$$\Psi_h = \Psi(x_1, \ldots, x_{d-1}, x_d \pm h(x_1, \ldots, x_{d-1}, x_{d+1}, \ldots, x_D), x_{d+1}, \ldots, x_D),$$

where $\Psi$ is the Heaviside function; $\Psi(x) = \mathbf{1}_{\{x \in I^D | x_d \geqslant 0\}}$. We define a set by the intersection of $J$ horizon functions $\Psi_{h_1}, \ldots, \Psi_{h_J}$; namely the family of sets is defined by

$$\mathcal{S}_{\alpha, J} := \left\{ S \subset [0, 1]^D \mid \mathbf{1}_S = \Psi_{h_1} \otimes \cdots \otimes \Psi_{h_J} \right\}.$$

Intuitively, $h$ is regarded as an $\alpha$-smooth curved surface in $I^D$, and $\Psi_h$ describes a set which is one side of the surface. Also, $S \in \mathcal{S}_{\alpha, J}$ is a set which is a intersection of $J$ sets by $\Psi_{h_1}, \ldots, \Psi_{h_J}$.

### A Support of Probability Measures

Let $N_x$ denote an open neighborhood of $x \in I^D$, and For a probability measure $P$, a support of $P$ is defined as

$$\mathrm{Supp}(P) := \left\{ x \in I^D \mid P(N_x) > 0, \forall N_x \in \Sigma \right\}.$$

## B  PROOFS

### B.1  PROOF OF LEMMA 1

Fix $m \in [M]$ and a corresponding $\widetilde{S}_m, S_m$ and $g_m$. For any $B \in \sigma(S_m)$, the definition of $\gamma_m$ yields

$$P_X(B) = P_Z(\gamma_m^{-1}(B)) = \int_{\gamma_m^{-1}(B)} p_Z(z) dz,$$

where $p_Z$ is a density function of a uniform measure $P_Z$. By changing variables $x = \gamma_m(z)$, we have

$$\int_{\gamma_m^{-1}(B)} p_Z(z) dz = \int_B p_Z(\gamma_m^{-1}(x)) J_{\gamma_m}(x) dx,$$

where $J_{\gamma_m}(x) = |\det \nabla g_m^{-1}(x)|$. Using $p_Z(z) = 1$ for all $z \in I^D$, we obtain the following form of $P_X(B)$ using a function $p_m : \widetilde{S}_m \to S_m$ as

$$P_X(B) = \int_B J_{g_m}(x) dx =: \int_B p_m(x) dx, \tag{8}$$

and $p_m$ is $\beta$-smooth since $\gamma_m$ is $\beta + 1$-smooth and bijective. $\qquad\square$

### B.2  PROOF OF LEMMA 2

Firstly, we show the first points. Suppose that $g$ is a continuous mapping. By the generalized intermediate value theorem (Theorem 24.3 in Munkres (2000)), we know that $g(I^D)$ is connected since $I^D$ is a connected set. Thus, a support of $P_g$ is connected. However, $P_g$ has a disconnected support, thus there is a contradiction. $\qquad\square$

### B.3 PROOF OF LEMMA 3

In this proof, $a \lesssim b$ denotes that $b$ is larger than $a$ up to a finite constant. $a \asymp b$ denotes that $a \lesssim b$ and $a \gtrsim b$ hold.

By the definition of $\{g_m\}_{m \in [M]}$ for the measure with local smoothness, we consider an explicit form of $g_m$. By Stein (2016), we can extend $g_m : \widetilde{S}_m \to S_m$ to $\widetilde{g}_m : I^D \to S_m$ since boundaries of $\widetilde{S}_m$ are Lipschitz continuous. Then, we provide the following formulation

$$\widetilde{g}_m(x) = (\gamma_{m,1}(x), ..., \gamma_{m,D}(x))^\top,$$

where $\gamma_{m,d} \in H^\beta(I^D)$. Then, we obtain the form of $g^*$ as

$$g^* = \sum_{m \in [M]} \widetilde{g}_m \otimes \mathbf{1}_{\widetilde{S}_m}.$$

Also, by the definition of $\mathcal{S}_{2\beta,J}$ which contains $\widetilde{S}_m$, we obtain the form

$$\mathbf{1}_{\widetilde{S}_m} = \bigotimes_{j \in [J]} \psi_{h_{m,j}},$$

with existing $\psi_{h_{m,j}}$. Then, we have

$$g^* = \sum_{m \in [M]} \widetilde{g}_m \bigotimes_{j \in [J]} \psi_{h_{m,j}}.$$

Preliminarily, we apply sub-neural networks from Yarotsky (2017) and Petersen & Voigtlaender (2017). Let $\zeta[\Theta_+]$ be a network for summation such that $\zeta[\Theta_+](x_1, ..., x_{D'}) = \sum_{d \in [D']} x_d$, and $\zeta[\Theta_\times]$ be a network for approximate multiplication such as $|\zeta[\Theta_\times](x_1, ..., x_{D'}) - \prod_{d \in [D']} x_d| < \epsilon$ with some $\epsilon > 0$ for all $x, x' \in I$ (Proposition 3 in Yarotsky (2017) and Lemma 1 in Imaizumi & Fukumizu (2018)).

We consider approximation $\sum_{m \in [M]} \gamma_{m,d} \bigotimes_{j \in [J]} \psi_{h_{m,j}}$ for each $d \in [D]$. Let $\zeta[\Theta_{\gamma,d,m}]$ and $\zeta[\Theta_{h,m,j}]$ for $d \in [D], m \in [M]$ and $j \in [J]$, and we will specify the networks later. Also, let $\zeta[\Theta_{S,m}] = \zeta[\Theta_\times](\zeta[\Theta_{h,m,1}](\cdot), ..., \zeta[\Theta_{h,m,1}](\cdot))$.

We consider a neural network

$$\zeta[\Theta_d] = \zeta[\Theta_+](\zeta[\Theta_\times](\zeta[\Theta_{\gamma,d,1}](\cdot), \zeta[\Theta_{S,1}](\cdot)), ..., \zeta[\Theta_\times](\zeta[\Theta_{\gamma,d,M}](\cdot), \zeta[\Theta_{S,M}](\cdot))).$$

Then, an approximation error is evaluated as

$$\left\| \sum_{m \in [M]} \gamma_{m,d} \bigotimes_{j \in [J]} \psi_{h_{m,j}} - \zeta[\Theta_d] \right\|_{L^2}$$

$$\leq \sum_{m \in [M]} \left\| \gamma_{m,d} \bigotimes_{j \in [J]} \psi_{h_{m,j}} - \zeta[\Theta_\times](\zeta[\Theta_{\gamma,d,m}](\cdot), \zeta[\Theta_{S,m}](\cdot)) \right\|_{L^2}$$

$$\leq \sum_{m \in [M]} \left\| \gamma_{m,d} \bigotimes_{j \in [J]} \psi_{h_{m,j}} - \zeta[\Theta_{\gamma,d,m}] \otimes \zeta[\Theta_{S,m}] \right\|_{L^2}$$

$$+ \sum_{m \in [M]} \left\| \zeta[\Theta_{\gamma,d,m}] \otimes \zeta[\Theta_{S,m}] - \zeta[\Theta_\times](\zeta[\Theta_{\gamma,d,m}](\cdot), \zeta[\Theta_{S,m}](\cdot)) \right\|_{L^2}$$

$$\leq \sum_{m \in [M]} \left\| \gamma_{m,d} \otimes \mathbf{1}_{\widetilde{S}_m} - \zeta[\Theta_{\gamma,d,m}] \otimes \zeta[\Theta_{S,m}] \right\|_{L^2} + M\epsilon_\times$$

$$\leq \sum_{m \in [M]} \left\| \gamma_{m,d} \otimes (\mathbf{1}_{\widetilde{S}_m} - \zeta[\Theta_{S,m}]) + (\gamma_{m,d} - \zeta[\Theta_{\gamma,d,m}]) \otimes \zeta[\Theta_{S,m}] \right\|_{L^2} + M\epsilon_\times$$

$$\leq \sum_{m \in [M]} \|\gamma_{m,d}\|_{L^\infty} \left\| \mathbf{1}_{\widetilde{S}_m} - \zeta[\Theta_{S,m}] \right\|_{L^2} + \|\gamma_{m,d} - \zeta[\Theta_{\gamma,d,m}]\|_{L^2} \|\zeta[\Theta_{S,m}]\|_{L^\infty} + M\epsilon_\times$$

$$=: \sum_{m \in [M]} T_{1,m} + T_{2,m} + M\epsilon_\times,$$

where the last inequality follows the Hölder's inequality.

About $T_{1,m}$, there exists a corresponding $\zeta[\Theta_{\gamma,d,m}]$ such that

$$\|\gamma_{m,d} - \zeta[\Theta_{\gamma,d,m}]\|_{L^2} \lesssim \|\Theta_{\gamma,d,m}\|_1^{-(\beta+1)/D},$$

by following Theorem A.8 in Petersen & Voigtlaender (2017). Also, since $\gamma_{m,d}$ is bounded by its smoothness and compact support, we have $\|\gamma_{m,d}\|_{L^\infty} < \infty$ hence

$$T_{1,m} \lesssim \|\Theta_{\gamma,d,m}\|_1^{-(\beta+1)/D}.$$

For $T_{2,m}$, we specify $\zeta[\Theta_{h,m,j}]$ as Theorem 3.1 in Petersen & Voigtlaender (2017). Then, we evaluate the following as

$$
\begin{aligned}
&\|\gamma_{m,d} - \zeta[\Theta_{\gamma,d,m}]\|_{L^2} \\
&\leqslant \left\| \bigotimes_{j\in[J]} \psi_{h_{m,j}} - \zeta[\Theta_\times](\zeta[\Theta_{h,m,1}](\cdot),...,\zeta[\Theta_{h,m,1}](\cdot)) \right\|_{L^2} \\
&\leqslant \left\| \bigotimes_{j\in[J]} \psi_{h_{m,j}} - \bigotimes_{j\in[J]} \zeta[\Theta_{h,m,j}] \right\|_{L^2} + \left\| \bigotimes_{j\in[J]} \psi_{h_{m,j}} - \zeta[\Theta_\times](\zeta[\Theta_{h,m,1}](\cdot),...,\zeta[\Theta_{h,m,1}](\cdot)) \right\|_{L^2} \\
&\lesssim \sum_{j\in[J]} \prod_{j'\in[J]\setminus\{j\}} \left( \left\|\psi_{h_{m,j'}}\right\|_{L^2} \vee \|\zeta[\Theta_{h,m,j'}]\|_{L^2} \right) \left\|\psi_{h_{m,j}} - \zeta[\Theta_{h,m,j}]\right\|_{L^2} + \epsilon_\times \\
&\leqslant \sum_{j\in[J]} \left\|\psi_{h_{m,j}} - \zeta[\Theta_{h,m,j}]\right\|_{L^2} + \epsilon_\times.
\end{aligned}
$$

Here, the last inequality follows the boundedness of $\psi_{h_{m,j'}}$ and $\zeta[\Theta_{h,m,j}]$ by Theorem 3.1 in Petersen & Voigtlaender (2017). Also, Theorem 3.1 in Petersen & Voigtlaender (2017) provides an existence of $[\Theta_{h,m,j}$ such that

$$\|\psi_{h_{m,j}} - \zeta[\Theta_{h,m,j}]\|_{L^2} \leqslant \|\Theta_{h,m,j}\|_1^{-\beta/(D-1)}.$$

We apply the boundedness of $\zeta[\Theta_{h,m,j}]$, we have

$$T_{2,m} \lesssim \sum_{j\in[J]} \|\Theta_{h,m,j}\|_1^{-\beta/(D-1)} + \epsilon_\times.$$

Combining the bounds for $T_{1,m}$ and $T_{2,m}$, we bound

$$
\begin{aligned}
&\left\| \sum_{m\in[M]} \gamma_{m,d} \bigotimes_{j\in[J]} \psi_{h_{m,j}} - \zeta[\Theta_d] \right\|_{L^2} \\
&\leqslant \sum_{m\in[M]} \|\Theta_{\gamma,d,m}\|_1^{-(\beta+1)/D} + \sum_{m\in[M]} \sum_{j\in[J]} \|\Theta_{h,m,j}\|_1^{-\beta/(D-1)} + (M+1)\epsilon_\times.
\end{aligned}
$$

Here, we consider a parameter $S = \sum_{d\in[D]} \|\Theta_d\|_1$ such that $S \asymp \|\Theta_{\gamma,d,m}\|_1 \asymp \|\Theta_{h,m,j}\|_1 \asymp \|\Theta_\times\|_1$. Also, following Yarotsky (2017) and Petersen & Voigtlaender (2017), a proper selection of $L = |\Theta_\times|$ and $B = \|\Theta_\times\|_\infty$ provides $\epsilon_\times \lesssim \|\Theta_\times\|_0^{-\beta/D}$. Then, we have

$$\left\| \sum_{m\in[M]} \gamma_{m,d} \bigotimes_{j\in[J]} \psi_{h_{m,j}} - \zeta[\Theta_d] \right\|_{L^2} \lesssim MJS^{-\beta/D}.$$

Since $J$ is finite, we obtain the result. $\qquad\square$

## B.4 Proof of Theorem 1

By the definition of $\hat{g}$ in (2), the following inequality holds

$$d_{\mathcal{F}}(P_n, P_{\hat{g},m}) = \sup_{f\in\mathcal{F}}(P_n f - P_{\hat{g},m} f) \leqslant d_{\mathcal{F}}(P_n, P_{g,m}) = \sup_{f\in\mathcal{F}}(P_n f - P_{g,m} f), \qquad (9)$$

for arbitrary $g \in \mathcal{G}$.

We consider a bound for $d_{\mathcal{F}}(P^*, P_{\hat{g}})$ as

$$
\begin{aligned}
d_{\mathcal{F}}(P^*, P_{\hat{g}}) &= \sup_{f \in \mathcal{F}}(P^* f - P_{\hat{g}} f) \\
&= \sup_{f \in \mathcal{F}}(P^* f - P_n f + P_n f - P_{\hat{g},m} f + P_{\hat{g},m} f - P_{\hat{g}} f) \\
&\leqslant \sup_{f \in \mathcal{F}}(P^* f - P_n f + P_{\hat{g},m} f - P_{\hat{g}} f) + \sup_{f \in \mathcal{F}}(P_n f - P_{\dot{g},m} f),
\end{aligned}
$$

where the inequality follows (9) with an existing $\dot{g} \in \mathcal{G}$. We will provide a detailed construction of $g^*$. We continue the bound as

$$
\begin{aligned}
&d_{\mathcal{F}}(P^*, P_{\hat{g}}) \\
&\leqslant \sup_{f \in \mathcal{F}}(P^* f - P_n f + P_{\hat{g},m} f - P_{\hat{g}} f) + \sup_{f \in \mathcal{F}}(P_n f - P^* f + P^* f - P_{\dot{g}} f + P_{\dot{g}} f - P_{\dot{g},m} f) \\
&\leqslant 2 \sup_{g \in \mathcal{G}} \sup_{f \in \mathcal{F}} |P_{g,m} f - P_g f| + \sup_{f \in \mathcal{F}}(P^* f - P_{\dot{g}} f) + 2 \sup_{f \in \mathcal{F}} |P_n f - P^* f| \\
&=: i + ii + iii.
\end{aligned}
$$

Here, $i$ denotes an effect from $m$ samplings, $ii$ denotes an approximation error, and $iii$ denotes an uncertainty with the $n$ observations.

To evaluate $i$ and $iii$, we provide the following lemma. This result follows a standard technique of the empirical process theory and we provide its outline for a sake of completeness.

**Lemma 4.** *Let $\mathcal{H}$ be a some set of measurable functions and $X_1, ..., X_n \sim P$ be i.i.d. $n$ observations. Suppose that $\mathbb{E}_P[h^2(X)] \leqslant \sigma^2$ and $\|h\|_{L^\infty} < C_h$ hold with finite parameters $\sigma^2 > 0$ and $C_h > 0$. Then, there exists a constant $C_\theta$ and we obtain*

$$
\begin{aligned}
&\sup_{h \in \mathcal{H}} \left| \frac{1}{n} \sum_{i \in [n]} h(X_i) - \mathbb{E}_P[h(X)] \right| \\
&\leqslant \inf_{\eta > 0} \left\{ 4\eta + 12 \int_\eta^{C_h} \sqrt{\frac{\log \mathcal{N}(\epsilon, \mathcal{H}, \|\cdot\|_{L^\infty})}{n}} d\epsilon + \sqrt{\frac{2\tau\sigma^2 + 4C_\theta}{n}} + \frac{\tau C_h}{n} \left( \frac{2}{3} + C_\theta \right) \right\}
\end{aligned}
$$

*with probability at least $1 - 2\exp(-\tau)$ for all $\tau > 0$.*

*Proof.* At the beginning, we bound an expectation of the empirical process (Steinwart & Christmann, 2008; Bartlett et al., 2005; Massart, 2000; Sriperumbudur et al., 2012). Afterward, we evaluate a concentration of the empirical process around the expectation.

By applying the symmetrization and concentration techniques (Proposition 7.10 in Steinwart & Christmann (2008)), we obtain

$$
\mathbb{E}_{P^{\otimes n}} \left[ \sup_{h \in \mathcal{H}} \left| \frac{1}{n} \sum_{i \in [n]} h(X_i) - \mathbb{E}[h(X)] \right| \right] \leqslant 2 \mathbb{E}_{P^{\otimes n} \otimes \nu^{\otimes n}} \left[ \sup_{h \in \mathcal{H}} \left| \frac{1}{n} \sum_{i \in [n]} u_i h(X_i) \right| \right],
$$

where $u_i \sim \nu$ is the Rademacher variable which takes 0 or 1 with probability 0.5. Combining this bound with the Taralgand's inequality (Theorem A.9.1 in Steinwart & Christmann (2008)), we obtain the following inequality

$$
\begin{aligned}
&\sup_{h \in \mathcal{H}} \left| \frac{1}{n} \sum_{i \in [n]} h(X_i) - \mathbb{E}_P[h(X)] \right| \\
&\leqslant (1 + \theta) \mathbb{E}_{P^{\otimes n} \otimes \nu^{\otimes n}} \left[ \sup_{h \in \mathcal{H}} \left| \frac{1}{n} \sum_{i \in [n]} u_i h(X_i) \right| \right] + \sqrt{\frac{2\tau\sigma^2}{n}} + \frac{\tau C_h}{n} \left( \frac{2}{3} + \frac{1}{\theta} \right), \qquad (10)
\end{aligned}
$$

with probability at least $1 - \exp(-\tau)$ for all $\tau > 0$ and $\theta > 0$.

About the term with the Rademacher variable, we also apply a similar strategy (Lemma A.4 in Bartlett et al. (2005)), then obtain

$$
\mathbb{E}_{P^{\otimes n} \otimes \nu^{\otimes n}} \left[ \sup_{h \in \mathcal{H}} \left| \frac{1}{n} \sum_{i \in [n]} u_i h(X_i) \right| \right] \leqslant \frac{1}{1 - \theta'} \mathbb{E}_{\nu^{\otimes n}} \left[ \sup_{h \in \mathcal{H}} \left| \frac{1}{n} \sum_{i \in [n]} u_i h(X_i) \right| \right] + \frac{\tau' C_h}{n \theta' (1 - \theta')},
\tag{11}
$$

with probability at least $1 - \exp(-\tau')$ for all $\tau' > 0$ and $\theta' > 0$.

Let $\| \cdot \|_n$ be an empirical norm as $\|f\|_n^2 = n^{-1} \sum_{i \in [n]} f(X_i)^2$. About the term $\mathbb{E}_{\nu^{\otimes n}} \left[ \sup_{h \in \mathcal{H}} \left| \frac{1}{n} \sum_{i \in [n]} u_i h(X_i) \right| \right]$, we apply the chaining technique and obtain

$$
\mathbb{E}_{\nu^{\otimes n}} \left[ \sup_{h \in \mathcal{H}} \left| \frac{1}{n} \sum_{i \in [n]} u_i h(X_i) \right| \right] \leqslant \inf_{\theta'' > 0} \left\{ 4 \theta'' + 12 \int_{\theta''}^{\tilde{C}_h} \sqrt{\frac{\log \mathcal{N}(\epsilon, \mathcal{H}, \| \cdot \|_n)}{n}} d\epsilon \right\}
$$

$$
\leqslant \inf_{\theta'' > 0} \left\{ 4 \theta'' + 12 \int_{\theta''}^{C_h} \sqrt{\frac{\log \mathcal{N}(\epsilon, \mathcal{H}, \| \cdot \|_{L^\infty})}{n}} d\epsilon \right\},
\tag{12}
$$

where the last inequality follows a bound for an empirical norm and the boundedness of $\mathcal{H}$.

Combining (10), (11) and (12) and changing variables, we obtain the result. □

To bound $I$ with $\widetilde{X}_1, ..., \widetilde{X}_m \sim P_g$, we consider the following value

$$
\sup_{g \in \mathcal{G}} \sup_{f \in \mathcal{F}} \left| \frac{1}{m} \sum_{i \in [m]} f(\widetilde{X}_i) - \mathbb{E}_{P_g}[h(X)] \right| = \sup_{g \in \mathcal{G}} \sup_{f \in \mathcal{F}} \left| \frac{1}{m} \sum_{i \in [m]} f \circ g(Z_i) - \mathbb{E}_{P_Z}[f \circ g(X)] \right|
$$

$$
=: \sup_{h \in \mathcal{H}} \left| \frac{1}{m} \sum_{i \in [m]} h(Z_i) - \mathbb{E}_{P_Z}[h(X)] \right|,
$$

where we define $\mathcal{H} = \{h = f \circ g \mid f \in \mathcal{F}, g \in \mathcal{G}\}$. To apply Lemma 4, we investigate a covering number $\mathcal{N}(\epsilon, \mathcal{H}, \| \cdot \|_{L^\infty})$.

**Lemma 5.** *Assume that $f \in \mathcal{F}$ is $L_1$-Lipschitz continuous. We obtain*

$$
\log \mathcal{N}(\epsilon, \mathcal{H}, \| \cdot \|_{L^\infty}) \leqslant \log \mathcal{N}((1 + L_1)^{-1} \epsilon, \mathcal{G}, \| \cdot \|_{L^\infty}) + \log \mathcal{N}((1 + L_1)^{-1} \epsilon, \mathcal{F}, \| \cdot \|_{L^\infty})
$$

*Proof.* Fix $\epsilon > 0$. Let $G \subset \mathcal{G}$ and $F \subset \mathcal{F}$ be covering sets as a set of centers of $\epsilon$-balls for the covering $\mathcal{G}$ and $\mathcal{F}$. Obviously, $|G| = \mathcal{N}(\epsilon, \mathcal{G}, \| \cdot \|_{L^\infty})$ and $|F| = \mathcal{N}(\epsilon, \mathcal{F}, \| \cdot \|_{L^\infty})$. We define a subset

$$
H := \left\{ h = f \circ g \mid g \in G, h \in F \right\} \subset \mathcal{H},
$$

and we known $|H| = |G| \times |F|$.

For any $h \in \mathcal{H}$, there exist $g \in \mathcal{G}$ and $f \in \mathcal{F}$, then $f = f \circ g$ holds. Also, by the definition of covering sets, there exist $f' \in F$ and $g' \in G$ such that $\|f - f'\|_{L^\infty} \leqslant \epsilon$ and $\|g - g'\|_{L^\infty} \leqslant \epsilon$. Let $h' = f' \circ g'$, and we measure the distance

$$
\|h - h'\|_{L^\infty} \leqslant \|f \circ g - f' \circ g'\|_{L^\infty}
$$

$$
= \|f \circ g - f \circ g' + f \circ g' - f' \circ g'\|_{L^\infty}
$$

$$
\leqslant \|f \circ g - f \circ g'\|_{L^\infty} + \|f \circ g' - f' \circ g'\|_{L^\infty}
$$

$$
\leqslant L_1 \|g - g'\|_{L^\infty} + \|f - f'\|_{L^\infty}
$$

$$
\leqslant (L_1 + 1)\epsilon.
$$

Here, the third inequality follows the Lipschitz property of $f \in \mathcal{F}$. Here, we know that $\mathcal{H}$ is covered by $(L_1 + 1)\epsilon$-balls with the center $H$. Since $|H| = |G| \times |F|$, the result holds. □

Now, we have the following entropy bound

$$\log \mathcal{N}(\epsilon, \mathcal{H}, \| \cdot \|_{L^\infty})$$
$$\leqslant \log \mathcal{N}((1 + L_1)^{-1}\epsilon, \mathcal{G}, \| \cdot \|_{L^\infty}) + \log \mathcal{N}((1 + L_1)^{-1}\epsilon, \mathcal{F}, \| \cdot \|_{L^\infty})$$
$$\leqslant (S_g + 1) \log(2(L_1 + 1)\epsilon^{-1}(L_g + 1)D_g B_g)$$
$$+ \min\left\{ (S_f + 1) \log(2(L_1 + 1)\epsilon^{-1}(L_f + 1)D_f B_f), C_\kappa(1 + L_1)\epsilon^{-\kappa} \right\}$$

with $D_g := \prod_{\ell \in [L_g + 1]}(D_\ell + 1)$ and $D_f := \prod_{\ell \in [L_f + 1]}(D_\ell + 1)$. Let $\Gamma_f := 2(L_f + 1)D_f B_f$, and $N_\epsilon(\widetilde{\mathcal{F}}) := \log \mathcal{N}(\epsilon, \widetilde{\mathcal{F}}, \| \cdot \|_n)$ for brevity. Here, we apply Lemma 8 in Schmidt-Hieber (2017) for the entropy bound for $\mathcal{G}$ and $\mathcal{F}$. Using the entropy bound and Lemma 4, we obtain

$$2 \sup_{f \in \mathcal{F}} |P_n f - P^* f|$$

$$\leqslant 4\eta + \frac{12}{n^{1/2}} \int_\eta^{C_{\widetilde{\mathcal{F}}}} \min\left\{ N_\epsilon(\widetilde{\mathcal{F}}), (S_f + 1)\log(\Gamma_f \epsilon^{-1}) \right\}^{1/2} d\epsilon$$

$$+ \frac{(2\tau\sigma^2 + C_\theta)^{1/2}}{n^{1/2}} + \frac{\tau C_h(2/3 + C_\theta)}{n}$$

$$\leqslant 4\eta + \frac{12}{n^{1/2}} \int_\eta^{C_{\widetilde{\mathcal{F}}}} N_\epsilon(\widetilde{\mathcal{F}})^{1/2} d\epsilon + \frac{12}{n^{1/2}}(S_f + 1)^{1/2}(\log^{1/2} \Gamma_f + C_{\widetilde{\mathcal{F}}}\log^{1/2} C_{\widetilde{\mathcal{F}}} - \eta \log^{1/2} \eta)$$

$$+ \frac{(2\tau\sigma^2 + C_\theta)^{1/2}}{n^{1/2}} + \frac{\tau C_h(2/3 + C_\theta)}{n}$$

$$\leqslant \Upsilon_n(\widetilde{\mathcal{F}}) + \frac{1}{n^{1/2}}\left( (S_f + 1)^{1/2}A_1 + A_2 \right) + \frac{A_3}{n}, \tag{13}$$

with some $\eta > 0$, $A_1 = 12 \log^{1/2} \Gamma_f$, $A_2 = C_{\widetilde{\mathcal{F}}}\log^{1/2} C_{\widetilde{\mathcal{F}}} + (2\tau\sigma^2 + C_\theta)^{1/2}$, and $A_3 = \tau C_h(2/3 + C_\theta)$. Also, we set $\Upsilon_n(\widetilde{\mathcal{F}}) = 4\eta + \frac{12}{n^{1/2}} \int_\eta^{C_{\widetilde{\mathcal{F}}}} N_{(L_1+1)^{-1}\epsilon}(\widetilde{\mathcal{F}})^{1/2} d\epsilon$. Then, we have

$$iii \leqslant \Upsilon_n(\widetilde{\mathcal{F}}) + \frac{1}{n^{1/2}}\left( (S_f + 1)^{1/2}A_1 + A_2 \right) + \frac{A_3}{n}. \tag{14}$$

About $I$, we define $\Gamma_g := 2(L_f + 1)D_g B_g$ and obtain a similar bound as

$$i \leqslant \Upsilon_m(\widetilde{\mathcal{F}}) + \frac{1}{m^{1/2}}\left( (S_g + 1)^{1/2}A_1' + A_2' \right) + \frac{A_3'}{m},$$

where $A_1' = 12(\log^{1/2} \Gamma_f + \log^{1/2} \Gamma_g)$, $A_2 = C_{\widetilde{\mathcal{F}}}\log^{1/2} C_{\widetilde{\mathcal{F}}} + C_{\mathcal{G}}\log^{1/2} C_{\mathcal{G}} + 2(2\tau\sigma^2 + C_\theta)^{1/2}$, and $A_3 = 2\tau C_h(2/3 + C_\theta)$.

About $ii$, we evaluate the error from approximation by constructing a specific deep neural network for generators. We apply Lemma 3 and let $\dot{g} = (\dot{g}_1, ..., \dot{g}_D)$ be a generator specified in Lemma 3.

$$ii = P_{g*}f - P_{\dot{g}}f$$

$$= \int (f \circ g^* - f \circ \dot{g}) dP_Z$$

$$\leqslant L_1 \int \|g^* - \dot{g}\|_2 dP_Z$$

$$\leqslant L_1 \left( \sum_{d \in [D]} \int |g_d^*(x) - \dot{g}_d(x)|^2 dP_Z(x) \right)^{1/2}$$

$$= L_1 \left( \sum_{d \in [D]} \|g_d^* - \dot{g}_d\|_{L^2}^2 \right)^{1/2}$$

$$\leqslant c_g L_1 D M S_g^{-\beta/D},$$

which follows $L_1$-Lipschitz continuity of $f$, the Jensen's inequality, the Cauchy-Schwartz inequality, compactness of the support $I^D$, and uniformity of $P_Z$. Then, we have

$$ii \leqslant c_2 M D S_g^{-\beta/D}.$$

Combining the result, we obtain the result of Theorem 1. □

### B.5 PROOF OF PROPOSITION 1

When $P$ are globally smooth, we obtain a $\beta$-smooth density function on $I^D$ by its definition. Due to the smoothness, the studies for nonparametric statistics (Nadaraya, 1964; Ghosal et al., 2007; Efromovich, 2010; van der Vaart & van Zanten, 2008; Tsybakov, 2009) guarantees that the methods (KDE,NB,SDE, and GP) obtain the rate $O(n^{-\beta/(2\beta+D)})$ with respect to the roof of $L^2$ norm.

When $P$ have disconnected supports and locally smooth, we consider a following specific $P$. Fix $M = 2$. Let us define supports as $S_1 = \widetilde{S}_1 = \{x \in I^D \mid x_1 \leqslant 0.5\} \subset I^D$ and $S_2 = \widetilde{S}_2 = \{x \in I^D \mid x_1 \leqslant 0.5\} \subset I^D$. Also, $g_1(z) = z$ and $g_2 : \widetilde{S}_2 \to S_2$ as

$$g_2(z) = (g_{2,1}(z_1), ..., g_{2,D}(z_D))^\top,$$

where $g_{2,1}(z_1) = 0.6 + c(z_1 - 0.5)^{1/3}$ and $g_{2,d}(z_d) = z_d$ for $d \in [D]\backslash\{1\}$ with a constant $c$. Then, by the proof of Lemma 1, $p_2(x)$ on $S_2$ is a quadratic function with respect to $z_1$ is 1-times differentiable but not twice-differentiable at the boundary $\{x \in I^D \mid x_1 = 0.6\}$. Hence, the studies (Nadaraya, 1964; Ghosal et al., 2007; Efromovich, 2010; van der Vaart & van Zanten, 2008; Tsybakov, 2009) provides that the generalization error of the methods is bounded by $O(n^{-\beta/(2\beta+D)})$ with $\beta = 1$. □

