# OpenReview forum: "Understanding GANs via Generalization Analysis for Disconnected Support"
_ICLR.cc/2019/Conference_

### Official Review · AnonReviewer1 · 2018-10-31
**Good stuff but comparison to other methods is overstated**

**Rating:** 6
**Confidence:** 3

**Review:**

This paper provides a theoretical study of GANs in the following setting:

- The target distribution has a locally smooth density on a compact set [0, 1]^D. It might be supported only on M disjoint components, each of which has a smooth boundary, within that compact set.
- The latent noise dimension (inputs to the generator) is of the same dimension as the data.
- An IPM loss (1) is used.
- The discriminator functions of the IPM and the generator networks are both ReLU networks of at most L layers and at most S total nonzero weights, with all weights having magnitude at most B.
- The discriminator functions are Lipschitz continuous; this is implied by the previous assumption, but the bound is tighter if we have a tighter constraint here.
- We obtain the generator which exactly minimizes the IPM between the empirical distributions of m samples from the model and n samples from the target.
- Maybe: there is some g such that P* is produced by g. It's not clear in your statement of Theorem 1 whether this is necessary, or exactly what it means when you say that in Lemma 3, and I haven't fully checked what's used yet; it would help to explicitly say this is not assumed if it's not.

Pointing out that the kinds of distributions handled by GANs often have disjoint support, and analyzing this case, is certainly of interest.

The full-dimensional support assumption is not ideal, but of course just because the analysis doesn't apply to most practical GAN settings doesn't mean it's not an important step towards one that does.

Also note that the assumption about the structure of the networks eliminates the MMD GANs that you use in experiments -- which have a kernel function at the top of the critic network -- though it does allow for most GAN variants. Maybe the most interesting algorithm for this setting is the Coulomb GAN (ICLR 2018, https://arxiv.org/abs/1708.08819 ), which uses a neural network critic of the kind you study but estimates a distance which (unlike the Wasserstein and most other GAN objective functions) has good statistical convergence properties (kappa=2 as you mention).

My biggest concern by far, though, is Proposition 1. You present it as if it's a lower bound: establishing that there is some class of distributions for which Theorem 1 shows that GANs can account for local smoothness and standard methods are shown not to be able to. This isn't what you do; instead you exhibit a class of distributions for which Theorem 1 shows that GANs can account for local smoothness, and previous analyses of standard methods do not show that they are able to take advantage of it. This is not the same thing at all! Although the previous upper bounds have matching lower bounds, you don't demonstrate (and it is likely not the case) that the distribution you show fits into the class of distributions used by the previous lower bounds, and so it remains very possible that other methods are able to take advantage of local smoothness as well as GANs do.

Given that Proposition 1 is hence a very weak statement, your main contribution in terms of disconnected support becomes "we can show that GANs can adapt to disconnected supports (under these other assumptions) that has not previously been shown for other methods." Not only is this a weaker result, but the degree to which you show GANs can take advantage of local smoothness is somewhat limited: at least with the parameter choices in Corollary 1, the smoothness only improves m dependence, not n dependence.

But in the GAN setting, m is essentially a question of how long you optimize for (and the relative rates between generator and discriminator updates, and various other questions like that out of scope for this paper), not any kind of externally fixed limitation like n. It's perhaps not too surprising that you don't show that optimizing a WGAN is statistically easier than estimating Wasserstein the distance, but given that estimating Wasserstein is so hard in high dimensions, it's a little disappointing. (Maybe it's easier, so that kappa is smaller, with locally smooth densities as here, though I don't know of any results like that offhand.)

(It's interesting, then, that in Corollary 1 the generator complexity depends only on m, basically the amount of optimization you're willing to do, while the discriminator complexity depends only on the available number of target samples n.)

Another concern is that to me, it is not very clear exactly what the statements of the assumptions mean. For example, does Theorem 1 apply only if I search over all generators in the class \mathcal G = \Xi(S_g, B_g, L_g)? In particular, does this mean that I have to consider all possible architectures matching those constraints, including allowing for all possible depths up to L_g, and all possible ways of allocating widths of the various interior layers / which weight entries are fixed at zero? It seems so, but this could be more explicit in the statements, even just by replacing "an existing \mathcal G" with "\mathcal G = \Xi(S_g, B_g, L_g)" in the statement of (e.g.) Theorem 1.

In your numerical experiments: you don't make it at all clear enough that you're plotting *different loss functions* for the GANs and the other methods! (You say this, but only in the text where it should definitely also be in the figure caption.) What happens if you plot the L2 difference for all methods, and the MMD/Wasserstein for all methods? Looking at Figure 6, it's not obvious that the GAN would do so substantially better. (It does seem to perhaps have the overall scale of the two components better than the other methods, but it doesn't look like as enormous a difference as it seems from Figure 5.)

Overall: Theorem 1 is of interest, but the results and especially the comparison to classic methods are not as resounding as they're presented here.

(Note that I have not (yet) verified or even really read most of the proofs; I might come back and do that later.)

Smaller points:

- I don't think that f-GAN actually fits in the framework (1) as you claim, since it needs to use the conjugate of f on the samples from Q. Also, the original GAN does fit into (1) but not your assumptions about the network form of f, since it needs a log in its activations.

- Another class of generative models where disconnected supports are really important is normalizing flows, which often build "bridges" between separated modes because (like your Lemma 2) their generators are constrained to be smooth and invertible. See e.g. Figure 2 of https://arxiv.org/abs/1810.01367 (who propose a new normalizing flow less susceptible to these problems).

- Remark 1 seems so obvious that it need not even be stated, since beta-smooth implies beta'-smooth for beta' < beta. It would only be interesting if you could actually take advantage of the smoother components somehow.

- Many papers in your bibliography are cited only as arXiv preprints when they were actually published in various places. For example, the first four papers were published at ICLR 2017, ICML 2017, ICML 2017, and ICLR 2018, respectively.

- There are many small typos and grammatical errors in the draft, including some that would be caught by a spell-checker ("methdos" on page 8), and an undefined LaTeX reference at the top of page 2. It would benefit from a thorough proofread.

---

> ### Author Response · Authors · 2018-11-17
> **Thank you for good comments. We will modify Proposition 1 and displays of the experiment.**
>
> > Proposition 1 and the following discussion presents it as if it's a lower bound
>
> In the proof of Proposition 1, we show that there exists a density function which has a first derivative but does not have a second derivative with disconnected support measures. Then, we discuss that the other standard methods fail to obtain a fast convergence rate with the densities without second derivatives. To clarify the point, we will update Proposition 1.
>
>
> > You don't show that optimizing a WGAN is statistically easier than estimating Wasserstein the distance, but given that estimating Wasserstein is so hard in high dimensions
>
> We agree that there are many kinds of practical difficulties for GANs including calculating Wasserstein divergence and optimizing parameters of generators and discriminators. Though these problems are significant, we cannot investigate all of them simultaneously, hence we focus on the generalization aspect in terms of sample size.
>
>
> > Does Theorem 1 apply only if I search over all generators in the class \mathcal G = \Xi(S_g, B_g, L_g)?
>
> We do not have to search over all generators since we partially specify the good DNN architecture for the bound. However, describing the partial specification for the architecture is quite complicated and it is not our main concern in our paper. Thus, we let Theorem 1 formally guarantee an existence of the good DNN architecture. The partial specification of the DNN architecture is discussed in our proof using a combination of network specification results by Yarotsky (2017), Schmidt-Hieber (2017), Petersen & Voigtlaender (2017), and so on.
>
>
> > You don't make it at all clear enough that you're plotting *different loss functions* for the GANs and the other methods
>
> Purpose of the experiment is NOT showing GANs perform better than others, BUT to show the settings about disconnectivity of supports affects performances of each of the methods. Regardless of the difference of measures, we can see that the disconnected support properties highly increase the error by the other methods by comparing Figure 5(A) and 5(B).
>
> > Looking at Figure 6, it's not obvious that the GAN would do so substantially better.
>
> An important point is that the other methods (except kernel (G)) provide estimators with larger supports than the true support due to the non-smoothness of the true density. Such estimation increases errors, hence it is a source of slow convergence of the other methods discussed in Figure 5. Though the kernel (G) method detect the true support, it is a result of overfitting inside the support, hence the overall generalization power is bad as shown in Figure 5.
>
> We will modify Figure 5 and 6 and make it more interpretable.

---

### Official Review · AnonReviewer2 · 2018-11-01
**Interesting analysis of the generalization performance of GANs, lacks strong experimental evidence**

**Rating:** 5
**Confidence:** 4

**Review:**

The paper provides some bounds on the generalization performance of GANs for approximating distributions with discontinuous support. This work relies heavily on the results shown in [1] and [2] on the approximation power of Deep networks for non-smooth functions. The paper is globally well written and the proof seems sound. However, the experiments could be more convincing and the relevance of the result is questionable:

- By choosing the function class F to the be L_1-lipschitz, the resulting error bound loses it’s dependence on the smoothness beta and becomes slightly worse than the classical methods (equation 7 with kappa = 2+2D). Is this an artifact of the proof? if that is the case, it would be good to have a tighter bound: [3] might be a good starting point.
- Neural networks used in practice are continuous usually, but it seems that all the analysis is all based on the fact that distributions with disjoint support require discontinuous networks. Can similar results be obtained in the more realistic case of continuous networks? Also what network architecture was used in the experiments?
- Although the bound in eq (5) clearly shows a tradeoff for S_g it only says that S_f should be as small as possible. Of course, if S_f =0 there is no discriminative power, but it’s unclear to me how the expression for S_f in eq (6)  can be obtained from (5) and why it would keep the discriminative power (in what sense?). Again, this tradeoff was discussed in prior work [3], so it might be worth looking into that direction.
- The discussion right after lemma 1 doesn't seem to be true: a distribution might have disjoint support and still have a density (i.e.: absolutely continuous with respect to the Lebesgue measure). It can even have a smooth density.
- The experiment doesn’t use the same metric to compare GANs method with other methods, so it is unclear how these methods compare. Moreover, figure 6 seems to show that other methods are also able to get the support right (Kernel E). Based on what could we claim that one method is better than the other?

Revision:
Thank you for your response.
> In fact, our estimator in the theoretical and experimental analysis employs a continuous (ReLU) network. Though discontinuous networks are necessary for our setting (Lemma 2), we show that (continuous) ReLU networks can approximate the discontinuous network effectively (Lemma 3), hence the effectiveness of GANs is proved (Theorem 1).

-That clarifies things, however I find that the discussion after lemma 2 rather missleading, if in the end the result ends up using continuous generator:
"Because of the discontinuity, generative models with smooth functions, such as an
adversarial generative model with kernel generators (Sinn & Rawat, 2018), cannot work well with
disconnected supports."

- It is still unclear to me how the optimal value of S_f is obtained from eq (5). The author points out the work by Zhang+ (2018), but this should be clarified in the current version of the paper: What result in Zhang+(2018) do you use to get this value?

- I find the experiments  not very convincing. I understand that the point is not to show that GANs are better than  other methods but it is important to be make meaningful compairisons (use comparable scores) otherwise there is little scientific value in figure 5 especially.

- As reviewer 1 mentions, lemma 3 is supposed to be one of  the main theoretical contributions of the paper, however, the proof seems very similar to the one in ([2], appendix B.1). Although the authors mention lemma 1 of [2] in the proof of lemma 3, it seems like the whole section in ([2] appendix B.1) is dedicated to show the very same result.

For all these reasons I still wouldn't recommend accepting this paper.






[1]: Yarotsky. Error bounds for approximation with deep relu networks.
[2]: Massaki Imaizumi, Kenji Fukumizu. Deep neural networks learn non-smooth functions effectively.
[3]: P. Zhang, Q. Liu, D. Zhou, T. Xu, and X. He. On the Discrimination-Generalization Tradeoff in GANs.

---

> ### Author Response · Authors · 2018-11-16
> **Thank you for good comments. We will modify some points and make the experimental evidence more interpretable.**
>
>
> > the resulting error bound loses it’s dependence on the smoothness beta and becomes slightly worse than the classical methods
>
> This is true, however, such the large discriminator (kappa = 2 + 2D) is not used in practice, hence we think it is not critical to discuss the actual performance of GANs.
>
>
> > it would be good to have a tighter bound: Zhang+ (2018) might be a good starting point.
>
> This point is not true. The result by Zhang+ (2018) (Corollary 3.2 in the paper) depends on a setting with a kappa = 2 case (they consider that discriminator is ReLU network with finite nodes in Corollary 3.2, hence kappa is always 2). Our setting can be regarded as a generalization of their setting.
>
>
> > Can similar results be obtained in the more realistic case of continuous networks? Also what network architecture was used in the experiments?
>
> In fact, our estimator in the theoretical and experimental analysis employs a continuous (ReLU) network. Though discontinuous networks are necessary for our setting (Lemma 2), we show that (continuous) ReLU networks can approximate the discontinuous network effectively (Lemma 3), hence the effectiveness of GANs is proved (Theorem 1).
>
>
> > how the expression for S_f in eq (6) can be obtained from (5) and why it would keep the discriminative power (in what sense?). It might be worth looking into that direction.
>
> S_f in (6) is the largest number of nodes which does not increase the generalization error, and the selection intuitively follows the discrimination analysis by Zhang+ (2018). Quantitative analysis for the discrimination power (e.g. deriving kappa in Zhang (2018)) is an interesting direction, hence we will try to clarify the point.
>
>
> > The discussion right after lemma 1 doesn't seem to be true.
>
> Your comment is right. We will correct the part to “we cannot obtain globally smooth densities.”
>
>
> > The experiment doesn’t use the same metric to compare GANs method with other methods, so it is unclear how these methods compare.
>
> Purpose of the experiment is NOT showing GANs perform better than others, BUT to show the settings about disconnectivity of supports affects performances of each of the methods. Regardless of the difference of measures, we can see that the disconnected support properties highly increase the error by the other methods by comparing Figure 5(A) and 5(B). We will modify Figure 5 and make it more interpretable.
>
>
> > figure 6 seems to show that other methods are also able to get the support right (Kernel E)
>
> We believe that you mention the kernel (G) method. The kernel (G) method detects the true support, however, it is a result of overfitting inside the support, hence the overall generalization power is bad as shown in Figure 5.

---

### Official Review · AnonReviewer3 · 2018-11-03
**Good efforts in approximation properties of deep generative networks; the results are weakly related to GANs**

**Rating:** 6
**Confidence:** 4

**Review:**

This paper claims three contributions.
1. We show that GANs perform better than other standard methods of estimating probability
measures when the measure satisfies the disconnected support property.

I prefer to state the contribution as "We show that deep generative networks perform better than other standard methods of estimating probability measures when the measure satisfies the disconnected support property." This claim is essentially from Lemma 3, and it is a property of deep generative networks instead of GANs. If we have another way to train deep generative networks (say, variational auto-encode), we still get the same good approximation error, just linearly dependent of the number of disconnected pieces. The proof of Lemma 3 is mainly from the definition of locally smoothness and the results in Petersen & Voigtlaender 2017. It's a nice effort to leverage the result in Petersen & Voigtlaender 2017 to prove approximation properties of deep generative models. The flaw of this part is that the claims of other standard methods (Proposition 1) is very hand-wavy and floppy. The proof of Proposition 1 has lots of typos. For example, the definition of S_1 and S_2. And this sentence "Then, by the proof of Lemma 1, p2(x) on S2 is a quadratic function with respect to z1 is 1-times differentiable but not twice-differentiable at the boundary ..." I guess that the authors want to argue that traditional function approximation methods (like Kernel methods, polynomial approximation) all have rate n^{-1/(2+D)} approximation rate in the L^2 norm when the function to approximate has discontinuities... However, the authors fail to make this point clear. Moreover, if we use a mixture model of traditional approximations, the rate will not be deteriorated.  And we will get similar results in Lemma 3. Then, even the claim "deep generative networks perform better than other standard methods of estimating probability measures when the measure satisfies the disconnected support property" is not that grounded.


2. We provide a new generalization error bound under a general formulation of GANs by analyzing an approximation error. The result is thus applicable to a wide range of variations of GANs.

This corresponding to the results in Theorem 1. The authors may want to write the assumption "all the discriminators are L_1 Lipschitz continuous" in the Theorem 1 explicitly, because this is an important assumption to get the results. The analysis in i & iii is standard. The analysis is the main contribution of this paper, and is from Lemma 3.

3. Based on the generalization bound, we provide a theoretical guideline for selecting architectures
of generators and discriminators.

I think the authors mean Equation (6) in this claim. However, this practical guidance is not practical, because (1) both \beta and \kappa are unknown in practice, especially \beta, (2) I can hardly image the number of connections (non-zero weights) will be my model design guidance instead of the model architecture.

In the numerical experiment, "We use d_F to evaluate errors by GANs, and a root of the expected squared errors with the L2-norm for the other methods." With different metrics, is this a fair comparison?

Finally, there are lots of typos in the paper and appendix. "refers to a probability measures", "The property makes a probability measure be complex", "with disconnected support", "Theorem 1, Corollary 1 and ??",

Among others, the original GAN(Goodfellow et al., 2014) is realized if F contains a logarithm of density ratio. The f-GAN (Nowozin et al., 2016) also belongs to this class. Equation (1) and (2) only includes Integral Probability Metrics, not divergences in Goodfellow et al., 2014 or Nowozin et al., 2016.

"smoothness and a dimension of data are sufficient to characterize an optimal convergence of generalization errors." If we allow mixture models, that's a different story.

"A boundary of S is J combination of -smooth hyper surfaces" Definition of S_{\alpha, J} is not clear. The exact definition in appendix is based on the definition of the horizon function, which include "x_d \pm h". Really confused about this \pm. The original definition in Petersen & Voigtlaender 2017 does not have this \pm.

“an ordinary density function cannot be defined.” Can an ordinary density function be defined by setting the value outside its support to be 0?

“an empirical norm \| \|” What is the empirical norm?

In Proof outline of Theorem 1, we have the decomposition of i, ii and iii. Should P_0 in iii be P^{*}?

"We compare the numerical performance of GANs and the other methods with toy data with."

More typos in the proofs, especially in proof of Lemma 3.

---

> ### Author Response · Authors · 2018-11-17
> **Thank you for good comments. We will modify Proposition 1 and displays of the experiment.**
>
> > If we have another way to train deep generative networks (say, variational auto-encode), we still get the same good approximation error
>
> Though we have the same guess such that variational autoencoders (VAEs) have the same approximation power, the generalization power of VAEs are basically unknown due to variational approximation due to another approximation techniques of VAEs. Hence, we put the methods including VAEs out of the scope of our paper. We consider that studying the generalization performance of the methods is an interesting research topic.
>
>
> > Proposition 1 fails to make the point.
>
> Thank you for your indication. We agree with your point and will update Proposition 1.
>
>
> > If we use a mixture model of traditional approximations, the rate will not be deteriorated.
>
> We think this point is not true. Firstly, it is difficult for to approximate a wide class of sets in a support (\mathcal{S}_{\alpha, J} in our paper) due to a shape restriction of components of mixtures (it yields a misspecification bias). Secondly, about infinite mixture models, we already discuss the disadvantage of the method as a nonparametric Bayesian method in Section 3.3.
>
>
> > The authors may want to write the assumption "all the discriminators are L_1 Lipschitz continuous" in the Theorem 1.
> > Typos
>
> Thank you for your point. We will modify the description.
>
>
> > This guidance for selection in (6) is not practical,
>
> It is a theoretical guideline, and it is not directly applied to practical use. We will add an explanation about the point in the updated paper.
>
>
> > With different metrics in the experiment, is this a fair comparison?
>
> Purpose of the experiment is NOT showing GANs perform better than others, BUT to show the settings about disconnectivity of supports affects performances of each of the methods. Regardless of the difference of measures, we can see that the disconnected support properties highly increase the error by the other methods by comparing Figure 5(A) and 5(B). We will modify Figure 5 and make it more interpretable.

---

> > ### Comment · AnonReviewer3 · 2018-12-12
> > **not all my concerns are addressed**
> >
> > I still prefer to state the contribution as "We show that deep generative networks perform better than other standard methods of estimating probability measures when the measure satisfies the disconnected support property." The claimed contribution on "generalization analysis" is misleading, because the novel new result is about approximation error.
> >
> > "Firstly, it is difficult for to approximate a wide class of sets in a support (\mathcal{S}_{\alpha, J} in our paper) due to a shape restriction of components of mixtures (it yields a misspecification bias)." I do not understand this reply. I still think that if we use a mixture model of traditional approximations, the theoretical rate will not be deteriorated.
> >
> > Their claim of the traditional method (Proposition 1) is still very floppy. The proof of Proposition 1 stays the same, with typos (definition of S_1 and S_2) and unclear derivations. The proof should be very rigorous, because it claims many well-known methods (tested by time) cannot do something.

---

### Meta-Review · Area_Chair1 · 2018-12-14
**Interesting theoretical results but should clarify contribution and improve math rigor**

**Confidence:** 5
**Recommendation:** Reject

**Metareview:**

This paper provides a theoretical analysis of GANs, showing its advantages when the measure satisfies the disconnected support property. Its main theoretical results are interesting, but the reviews and discussion shows some misleading places.  It was also found some of the claims and proof are not mathematically rigorous.